# Adversarial-Enhanced Causal Multi-Task Framework for Debiasing Post-Click Conversion Rate Estimation

## ABSTRACT

In real-world industrial scenarios, post-click conversion rate (CVR) prediction models are trained offline based on click events and subsequently applied online to both clicked and unclicked events. Unfortunately, unclicked events are inevitably difficult to estimate due to user self-selection, which leads to a degradation of CVR prediction accuracy. In order to estimate the prediction of unclicked events, the current mainstream Doubly Robust (DR) estimators introduce the concept of imputed errors. However, inaccuracies in imputed errors can increase the uncertainty in the generalization bound of CVR predictions, consequently resulting in a decline in the CVR prediction accuracy. To challenge this issue, we first present a theoretical analysis of the bias and variance inherent in DR estimators and then introduce a novel causal estimator that seeks to strike a balance between bias and variance within the DR framework, thus optimizing the learning of the imputation model in a more robust manner. Additionally, drawing inspiration from adversarial learning techniques, we propose a novel dual adversarial component, which learns from both the space level and the task level to eliminate the causal influence of input features on the CTR task (i.e., the click propensity), with the goal of achieving unbiased estimations. Our extensive experimental evaluations, conducted on both the widely used benchmark and the real-world large-scale Internet giant platform, convincingly demonstrate the effectiveness of our proposed scheme. Besides, we aim to release a high-quality dataset used for selection bias research in the advertising field.

## CCS CONCEPTS

• **Information systems → Recommender systems**.

## KEYWORDS

Recommender Systems; Post-click Conversion Rate; Selection Bias

**ACM Reference Format:**
Anonymous Author(s). 2024. Adversarial-Enhanced Causal Multi-Task Framework for Debiasing Post-Click Conversion Rate Estimation. In *Proceedings of The International World Wide Web Conference (WWW '24)*. ACM, New York, NY, USA, 9 pages. https://doi.org/XXXXXXX.XXXXXXX

## 1 INTRODUCTION

Affiliate advertising [17, 18, 29] is a major player in Internet advertising, and accurately predicting the post-click conversion rate (CVR)

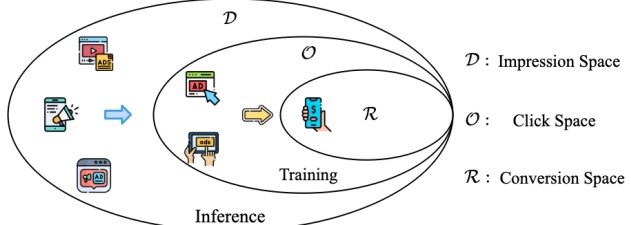

**Figure 1: Illustration of data sparsity and sample selection bias in the CVR task, where the training space comprises only clicked events, while the inference space encompasses all impression events.**

is essential for the revenue growth of affiliate advertising. Post-click conversion rate, which represents the likelihood of an advertisement generating payments following a click, serves as a pivotal metric for evaluating the effectiveness of advertising campaigns and is a major contributor to boost the gross merchandise volume (GMV) of the platform. However, the post-click conversion feedback can only be observed in the click space. Hence, conventional methods [11, 19] often concentrate on training CVR tasks based on click events and then predicting all impression events, which inevitably encounters two critical challenges: (1) Data Sparsity: The conversion labels are extremely sparse in real-world applications. For instance, the Ali-CCP dataset [12] indicates that approximately 3.75% of the exposed samples were clicked, with only about 0.025% of those samples resulting in conversions. (2) Sample Selection Bias: The presence of the clicked events is influenced by the user's propensity, which means that the clicks and conversions are causal and the missing conversion labels are missing-not-at-random [2, 12, 16], leading to a misalignment of event distributions between the training space and the inference space, as illustrated in Fig. 1.

To challenge these two issues, researchers have explored modeling the CVR task within the impression space [12, 25]. Unfortunately, this solution overlooks the essential causal relationship between clicks and conversions, thus leading to a biased CVR prediction. Currently, most of the debiasing methods utilize the doubly robust (DR) estimator [6, 15, 26, 28] to further correct biased estimates in a doubly robust manner. The unbiasedness of this estimator hinges on the accuracy of either the imputed errors or the propensity scores. In general, the DR estimator has exhibited superior performance for debiasing post-click conversion rate estimation [24]. Nevertheless, two potential challenges undermine its double robustness:

- **Potential Inaccurate Deviation (PID):** This challenge arises when error deviations are inaccurate, leading to a significant increase in the estimator's bias, variance, and generalization bounds, as discussed in our theoretical analysis in Section 3.1 and Section 3.2. This can potentially result in unfavorable outcomes and reduced reliability.

- **Limited Debias Capacity (LDC):** In unclicked events, the inverse of the propensity score tends to be large. Error deviations in the imputation model are susceptible to inaccuracy due to selection bias and data sparsity. Consequently, relying solely on the objective function for corrective action may be naive and limited in its effectiveness.

In this paper, we propose an Adversarial-Enhanced Causal Multi-task framework dubbed AECM to tackle the above-mentioned issues. To deal with the PID problem, we first derive a thorough analysis of the bias and variance associated with DR estimators, as elaborated in section 3.1. Building on this analysis, we then redefine the objective of the imputation task with the aim of minimizing both bias and variance. To achieve this goal, we propose a novel estimator designed to jointly optimize both bias and variance. This novel estimator is a key component of the AECM framework, facilitating precise prediction of imputed errors and ensuring unbiased CVR prediction. For a more comprehensive understanding of our proposed estimator, please refer to section 3.2 for details.

To deal with the LDC problem, we introduce a novel dual adversarial module comprising both space-level and task-level adversarial components. For the space-level adversarial module, it eliminates the influence of input features on the Click-Through Rate (CTR) task (i.e., click propensity) by performing a min-max game to align the distributions of clicked and unclicked events in the impression space, thus obtaining conditionally unbiased embeddings on the space level. For the task-level adversarial module, it is utilized to denoise the click propensity for the CVR task. Technically, on one hand, we introduce a discriminator to learn task-invariant features (i.e., click-related information) for both CTR and CVR. To ensure unbiased CVR embeddings, we incorporate an orthogonal loss that encourages dissimilarity with task-invariant features, ultimately achieving an unbiased embedding free from click propensities. On the other hand, we assume that all events in the impression space are clicked and conduct the task-level adversarial learning in a counterfactual manner. Thus, we establish a generator to align the distribution of fake embeddings generated by the CTR task with the real CVR embeddings. Extensive experiments are conducted to demonstrate the superiority of our proposed framework compared with state-of-the-art techniques.

The contributions of this paper can be summarized as follows:

- We propose a novel adversarial-enhanced causal multi-task framework to deal with the selection bias issue for post-click conversion rate estimation. Our framework seamlessly integrates adversarial learning and causal multi-task learning.
- We present a theoretical analysis of the bias and variance inherent in DR estimators and design a novel causal estimator to effectively minimize both bias and variance from the bias-variance trade-off perspective.
- We present a novel dual adversarial module, operating on both the space and task level. It can denoise the click propensity and capture counterfactual embeddings to optimize the CVR task.
- We aim to release a high-quality large-scale dataset used for selection bias research in the advertising field. We hope our datasets can serve as a benchmark to facilitate the research of selection bias in the Recommender Systems.

**Table 1: Important notations and descriptions.**

| Symbol | Description |
| :---: | :---: |
| $O$ | the click space |
| $\mathcal{D}$ | the impression space |
| $o_{u,i}$ | the ground-truth label of CTR |
| $\hat{p}_{u,i}$ | the predicted propensity of CTR |
| $r_{u,i}$ | the ground-truth label of CVR |
| $\hat{r}_{u,i}$ | the predicted propensity of CVR |
| $\hat{t}_{u,i}$ | the predicted propensity of CTCVR |
| $\hat{e}_{u,i}$ | the predicted value of CVR loss |
| $\delta_{u,i}$ | the difference between $e_{u,i}$ and $\hat{e}_{u,i}$ |

## 2 PRELIMINARIES

### 2.1 Problem Formulation

Let $\mathcal{U} = \{u_1, u_2, \cdots, u_m\}$ be a set of $m$ users, $\mathcal{I} = \{i_1, i_2, \cdots, i_n\}$ be the set of $n$ items, and $\mathcal{D} = \mathcal{U} \times \mathcal{I}$ be the set of all user-item pairs over the impression space. Denote $O \in \{0, 1\}^{m \times n}$ as the click matrix where each entry $o_{u,i}$ indicates whether a click action takes place between user $u$ and item $i$, $\mathcal{R} \in \{0, 1\}^{m \times n}$ as the conversion matrix where each entry $r_{u,i}$ indicates whether a conversion action occurs after user $u$ clicks item $i$. If we have a fully observed conversion matrix $\mathcal{R}$, the ideal loss function can be formulated as:

$$\mathcal{L}_{ideal}(\mathcal{R}, \hat{\mathcal{R}}) = \frac{1}{|\mathcal{D}|} \sum_{(u,i) \in \mathcal{D}} e(r_{u,i}, \hat{r}_{u,i}), \quad (1)$$

where $\hat{\mathcal{R}}$ denotes the predicted conversion matrix, $\hat{r}_{u,i}$ denotes the predicted conversion label and $e$ represents the prediction error.

In reality, only post-click conversions can be observed, i.e., $r_{u,i}$ only exists in the click space $O$. Hence, the naive estimator estimates the ideal loss $\mathcal{L}_{ideal}$ for clicked events as

$$\mathcal{L}_{naive}(\mathcal{R}, \hat{\mathcal{R}}) = \frac{1}{|O|} \sum_{(u,i) \in \mathcal{D}} o_{u,i} e(r_{u,i}, \hat{r}_{u,i}). \quad (2)$$

However, due to the selection bias, the conversion labels for unclicked events are missing not at random, which leads to a biased estimation for the widely adopted naive estimator, i.e., $\mathbb{E}_O[\mathcal{L}_{naive}] \neq \mathcal{L}_{ideal}$. In the following, we will briefly outline four typical and latest unbiased estimators.

### 2.2 Existing Estimators

**Inverse Propensity Weighting Estimator.** In order to eliminate the distribution error between the click space and the impression space, the inverse propensity weighting (IPW) estimator [1, 16, 28] weights each clicked event with $1/p_{u,i}$, where $p_{u,i} = \mathbb{E}[o_{u,i}]$ indicates the propensity of user $u$ clicking on item $i$. The IPW estimator can be formulated as

$$\mathcal{L}_{IPW}(\mathcal{R}, \hat{\mathcal{R}}) = \frac{1}{|O|} \sum_{(u,i) \in \mathcal{D}} \frac{o_{u,i} e(r_{u,i}, \hat{r}_{u,i})}{\hat{p}_{u,i}}. \quad (3)$$

**Doubly Robust Estimator.** The IPW estimator derives an unbiased estimate on the condition of accurate prediction for any click event, which is difficult to guarantee. Therefore Doubly Robust (DR) estimator [6, 15, 26, 28] introduces an error imputation task to additionally model the prediction error of conversions $\hat{e}(r_{u,i}, \hat{r}_{u,i})$ in $\mathcal{D}$, which corrects the estimates in a doubly robust way. The DR

estimator can be formulated as

$$\mathcal{L}_{DR}(\mathcal{R}, \hat{\mathcal{R}}) = \frac{1}{|\mathcal{D}|} \sum_{(u,i) \in \mathcal{D}} \hat{e}(r_{u,i}, \hat{r}_{u,i}) + \frac{o_{u,i}\delta_{u,i}}{\hat{p}_{u,i}}, \quad (4)$$

where $\delta_{u,i} = e(r_{u,i}, \hat{r}_{u,i}) - \hat{e}(r_{u,i}, \hat{r}_{u,i})$ denotes the error deviation.

**Doubly Robust Joint Learning Estimator.** In addition to employing $\mathcal{L}_{DR}$ to learn the conversion matrix, Doubly Robust Joint Learning (DR-JL) estimator [24] trains the imputation model by minimizing the squared deviations of the imputed errors from the prediction errors:

$$\mathcal{L}_e^{DR-JL}(\mathcal{R}, \hat{\mathcal{R}}) = \frac{1}{|\mathcal{O}|} \sum_{(u,i) \in \mathcal{O}} \frac{\delta_{u,i}^2}{\hat{p}_{u,i}}. \quad (5)$$

**More Robust Doubly Robust Estimator.** DR estimator suffers from the high variance issue, in order to mitigate this and obtain a more robust estimator, More Robust Doubly Robust estimator (MRDR) [8] proposes $\mathcal{L}_e^{MRDR}$ to learn the imputed error with the following loss function

$$\mathcal{L}_e^{MRDR}(\mathcal{R}, \hat{\mathcal{R}}) = \frac{1}{|\mathcal{O}|} \sum_{(u,i) \in \mathcal{O}} \frac{1 - \hat{p}_{u,i}}{\hat{p}_{u,i}} \cdot \frac{\delta_{u,i}^2}{\hat{p}_{u,i}}. \quad (6)$$

## 3 PROPOSED METHOD

In order to deal with the challenges posed by selection bias and data sparsity, we propose a novel Adversarial-Enhanced Causal Multi-task framework dubbed AECM, as illustrated in Fig. 2. In the following subsections, we provide a novel bias-variance trade-off estimator, along with a comprehensive explanation of the dual adversarial components, encompassing both the space-level and the task-level adversarial modules.

### 3.1 Bias and Variance Analysis of DR Estimator

Initially, we formulate the bias and variance of DR estimator.

**THEOREM 1.** *Let $\delta_{u,i} = e_{u,i} - \hat{e}_{u,i}$ denote the error deviation and $\Delta_{u,i} = \frac{p_{u,i} - \hat{p}_{u,i}}{\hat{p}_{u,i}}$ denote multiplicative propensity deviation. The bias of the DR estimator is*

$$\text{Bias}\left[\mathcal{L}_{DR}(\mathcal{R}, \hat{\mathcal{R}})\right] = \frac{1}{|\mathcal{D}|} \left| \sum_{(u,i) \in D} \Delta_{u,i}\delta_{u,i} \right|. \quad (7)$$

PROOF. For a single term of the DR estimator, its bias on the click indicator $o_{u,i}$ is

$$\text{Bias}\left[\mathcal{L}_{DR}(\mathcal{R}, \hat{\mathcal{R}})\right] = \left| \mathbb{E}_O\left[\mathcal{L}_{DR}(\mathcal{R}, \hat{\mathcal{R}})\right] - \mathcal{L}_{\text{ideal}}(\mathcal{R}, \hat{\mathcal{R}}) \right|$$

$$= \left| \frac{1}{|\mathcal{D}|} \sum_{(u,i) \in \mathcal{D}} \mathbb{E}_O\left[\hat{e}_{u,i} + \frac{o_{u,i}\delta_{u,i}}{\hat{p}_{u,i}} - e_{u,i}\right] \right|$$

$$= \left| \frac{1}{|\mathcal{D}|} \sum_{(u,i) \in \mathcal{D}} \left[\hat{e}_{u,i} + \frac{p_{u,i}\delta_{u,i}}{\hat{p}_{u,i}} - e_{u,i}\right] \right|$$

$$= \frac{1}{|\mathcal{D}|} \left| \sum_{(u,i) \in D} \Delta_{u,i}\delta_{u,i} \right|.$$

**THEOREM 2.** *Let $\delta_{u,i} = e_{u,i} - \hat{e}_{u,i}$ denote the error deviation and the variance of the DR estimator is*

$$\mathbb{V}_O\left[\mathcal{L}_{DR}(\mathcal{R}, \hat{\mathcal{R}})\right] = \frac{1}{|\mathcal{D}|^2} \sum_{(u,i) \in \mathcal{D}} \frac{p_{u,i}\left(1 - p_{u,i}\right)}{\hat{p}_{u,i}^2} \delta_{u,i}^2. \quad (8)$$

PROOF. For a single term of the DR estimator, its variance on the click indicator $o_{u,i}$ is

$$\mathbb{V}_O\left[\mathcal{L}_{DR}(\mathcal{R}, \hat{\mathcal{R}})\right] = \mathbb{V}_O\left[\frac{1}{|\mathcal{D}|} \sum_{(u,i) \in \mathcal{D}} \hat{e}_{u,i} + \frac{o_{u,i}\delta_{u,i}}{\hat{p}_{u,i}}\right]$$

$$= \frac{1}{|\mathcal{D}|^2} \sum_{(u,i) \in \mathcal{D}} \mathbb{V}_{o_{u,i}}\left[\hat{e}_{u,i} + \frac{o_{u,i}\delta_{u,i}}{\hat{p}_{u,i}}\right]$$

$$= \frac{1}{|\mathcal{D}|^2} \sum_{(u,i) \in \mathcal{D}} \frac{p_{u,i}\left(1 - p_{u,i}\right)}{\hat{p}_{u,i}^2} \delta_{u,i}^2.$$

Theorem 1 and Theorem 2 illustrate that the bias and variance of DR estimators depend on the click propensity $\hat{p}_{u,i}$, and the error deviation $\delta_{u,i}$. Inaccurate estimations of either $\hat{p}_{u,i}$ or $\delta_{u,i}$ will result in a high bias and variance problem. Therefore, a robust estimator should guarantee that both $\hat{p}_{u,i}$ and $\delta_{u,i}$ are accurately predicted.

### 3.2 Bias-Variance Trade-off Estimator

The following Theorem 3 also underscores the importance of accurately estimating $\hat{p}_{u,i}$ and $\delta_{u,i}$ from another perspective. Given the observed conversion matrix $\mathcal{R}^o$, the optimal prediction matrix $\hat{\mathcal{R}}^\dagger$ [20] is obtained by minimizing the estimated prediction inaccuracy using the DR estimator over a hypothesis space $\mathcal{H}$:

$$\hat{\mathcal{R}}^\dagger = \underset{\hat{\mathcal{R}} \in \mathcal{H}}{\text{argmin}} \left\{ \mathcal{L}_{DR}\left(\mathcal{R}^o, \hat{\mathcal{R}}\right) \right\} \quad (9)$$

The prediction inaccuracy of the optimal prediction matrix has the following generalization bound [24].

**THEOREM 3.** *(Generalization Bound). For any finite hypothesis space $\mathcal{H}$ of prediction matrices, with probability $1 - \eta$, the prediction inaccuracy of the optimal prediction matrix using the DR estimator has the upper bound*

$$\mathcal{L}_{DR}\left(\mathcal{R}^o, \hat{\mathcal{R}}^\dagger\right) + \underbrace{\sum_{(u,i) \in \mathcal{D}} \frac{\left|\Delta_{u,i}\delta_{u,i}^\dagger\right|}{|\mathcal{D}|}}_{\textit{Bias Term}} + \underbrace{\sqrt{\frac{\log\left(\frac{2|\mathcal{H}|}{\eta}\right)}{2|\mathcal{D}|^2} \sum_{(u,i) \in \mathcal{D}} \left(\frac{\delta_{u,i}^\ddagger}{\hat{p}_{u,i}}\right)^2}}_{\textit{Variance Term}},$$

*where $\Delta_{u,i} = \frac{p_{u,i}(1-p_{u,i})}{\hat{p}_{u,i}^2}$, $\delta_{u,i}^\ddagger$ denotes the error deviation corresponding to the prediction matrix $\mathcal{R}^\ddagger = \text{argmax}_{\hat{\mathcal{R}}^h \in \mathcal{H}} \left\{ \sum_{(u,i) \in \mathcal{D}} \left(\delta_{u,i}^h / \hat{p}_{u,i}\right)^2 \right\}$*

The generalization bound illustrated in Theorem 3 comprises a bias term and a variance term, both of which also exhibit a correlation with the magnitude of the click propensity $\hat{p}_{u,i}$ and the error deviation $\delta_{u,i}$. Therefore, these two factors with larger biases can increase the uncertainty in the generalization upper bound of CVR predictions, subsequently resulting in a decline in the model's prediction accuracy.

Expanding upon the previously discussed insights, we delve into the development of a novel estimator, which aims to bolster the

**Figure 2: The model architecture of AECM, comprises the dual adversarial components (i.e., the space-level adversarial module and the task-level adversarial module), along with a bias-variance trade-off estimator. These elements work in tandem to ensure unbiased estimation, contributing from adversarial learning and causal learning perspectives, respectively.**

robustness of sample selection bias (SSB) correction at the loss level. As outlined in Section 2.2, the MRDR estimator attempts to directly minimize the variance of $\mathcal{L}_{DR}(\mathcal{R}, \hat{\mathcal{R}})$, which is expected to yield a more resilient performance. Nevertheless, it is essential to emphasize that this strategy is particularly effective when the bias is relatively small, as indicated by the insights derived from Theorem 3. In cases where $Bias\left[\mathcal{L}_{DR}(\mathcal{R}, \hat{\mathcal{R}})\right]$ is substantial, the efficacy of variance reduction diminishes. Drawing inspiration from this observation, we formulate the following optimization expression for the imputation task, considering the bias-variance trade-off to adaptively control the generalization bound:

$$
\begin{aligned}
\mathcal{L}_e^{Balance}(\lambda_\tau) = & \frac{1}{|O|} \sum_{(u,i) \in O} \lambda_\tau \cdot \frac{(1 - \hat{p}_{u,i})^2}{\hat{p}_{u,i}^2} \cdot \frac{\delta_{u,i}^2}{\hat{p}_{u,i}} \\
& + \frac{1}{|O|} \sum_{(u,i) \in O} (1 - \lambda_\tau) \cdot \frac{1 - \hat{p}_{u,i}}{\hat{p}_{u,i}} \cdot \frac{\delta_{u,i}^2}{\hat{p}_{u,i}},
\end{aligned}
\tag{10}
$$

where the preceding term and the subsequent term correspond to the bias and variance parts discussed in Section 3.1, respectively.

Besides, by incorporating the weighting factor $\lambda_\tau$, we establish an effective equilibrium between the bias and variance of the DR estimator, thereby facilitating a more harmonious trade-off. This adjustment also contributes to heightened stability in the upper bound of the generalization bound. In addition, upon comparing the loss function of imputation learning in (10) with the one in equation (4), a discernible modification comes to light. In (10), the weight term $\frac{1}{\hat{p}_{u,i}}$ is substituted with $\frac{1-\hat{p}_{u,i}}{\hat{p}_{u,i}^2}$, which possesses specific properties that are of significance:

$$
\begin{cases}
\frac{1}{\hat{p}_{u,i}} < \frac{1-\hat{p}_{u,i}}{\hat{p}_{u,i}^2}, & \text{if } \hat{p}_{u,i} < \frac{1}{2} \\
\frac{1}{\hat{p}_{u,i}} > \frac{1-\hat{p}_{u,i}}{\hat{p}_{u,i}^2}, & \text{if } \hat{p}_{u,i} > \frac{1}{2}.
\end{cases}
\tag{11}
$$

Consequently, the balance estimator functions by augmenting the penalty for clicked events with low propensity while concurrently diminishing the penalty for the remaining events. This strategic adjustment incentivizes the imputation task to allocate more attention to the inaccurate aspects of the propensity score model. In summary, the CVR loss can be expressed as follows:

$$
\begin{aligned}
\mathcal{L}_{CVR} &= \mathcal{L}_{DR}(\mathcal{R}, \hat{\mathcal{R}}) + \mathcal{L}_e^{Balance}(\lambda_\tau) \\
&= \frac{1}{|\mathcal{D}|} \sum_{(u,i) \in \mathcal{D}} \hat{e}(r_{u,i}, \hat{r}_{u,i}) + \frac{o_{u,i} \delta_{u,i}}{\hat{p}_{u,i}} + \mathcal{L}_e^{Balance}(\lambda_\tau),
\end{aligned}
\tag{12}
$$

where $\mathcal{L}_{DR}(\mathcal{R}, \hat{\mathcal{R}})$ and $\mathcal{L}_e^{Balance}(\lambda_\tau)$ represent the objective function of CVR tower and imputation tower, respectively.

## 3.3 Space-Level Adversarial Module

In practical, user interactions with advertisements are influenced by a multitude of factors, which encompass diverse exposure styles and ad positions [30]. In essence, the events observed in the click space are subject to the user's propensity, whereby events associated with a higher propensity exhibit a greater likelihood of being incorporated into the click space. This phenomenon engenders the challenge of biased estimation, stemming from SSB. It is noteworthy that advertisements characterized by low click propensity, despite their absence in the observed click space, possess the potential for conversions if hypothetically clicked. Consequently, the SSB dilemma leads to a situation where unclicked events cannot receive effective training, directly resulting in a distribution inconsistency between the training and inference spaces.

Hence, prevailing propensity-based causal correction techniques, including the Inverse Propensity Weighting Estimator (IPW) and the Doubly Robust Estimator (DR), are designed to mitigate the causal impact of input features on the CTR task, as illustrated in

  

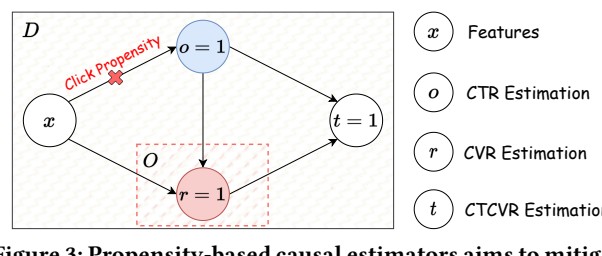

**Figure 3: Propensity-based causal estimators aims to mitigate the causal impact of input features on the CTR task.**

Fig. 3. In contrast, our proposed AECM extends these methodologies by integrating adversarial learning techniques [4, 5, 9, 27] to confront this challenge more effectively. Specifically, we employ a click discriminator tasked with classifying the domain label of each event (i.e., whether it belongs to the clicked or unclicked category) based on its representation from the embedding layer. The basic assumption is that if the click discriminator encounters challenges in accurately predicting an event's domain label, it signifies that the event's representation has effectively removed the click propensity information. Furthermore, we incorporate an auxiliary loss to confound the click discriminator and maximize the domain classification loss. Concurrently, the click discriminator endeavors to minimize the domain classification loss, thereby striving to become a robust classifier. The formal formulations are provided below:

$$\mathcal{L}_{Space} = \ell_d + \ell_g, \tag{13}$$

$$\ell_d = \mathbb{E}_{p(z_1)}\left[\left(\mathcal{D}(x_{clk}) - a_1\right)^2\right] + \mathbb{E}_{p(z_2)}\left[\left(\mathcal{D}(x_{unclk}) - b_1\right)^2\right] \tag{14}$$

$$\ell_g = \mathbb{E}_{p(z_2)}\left[\left(\mathcal{D}(x_{unclk}) - a_1\right)^2\right], \tag{15}$$

where $x_{unclk}$ and $x_{clk}$ denote the representations of unclicked events and clicked events respectively, $a_1$ and $b_1$ represent their domain labels. $p(z_1)$ and $p(z_2)$ represent data distributions. $\mathcal{D}$ corresponds to the click discriminator.

The functions delineated above are devised with the primary objective of achieving a state where the representations of clicked and unclicked advertisements become indistinguishable. Consequently, this alignment of learned representations from both domains facilitates the acquisition of unbiased embeddings that are entirely devoid of any click propensity information.

### 3.4 Task-Level Adversarial Module

Despite the fact that the potential alleviation of biased CVR estimation through the removal of click propensity, it is essential to recognize that the task is typically constrained to training within the click space. However, the analysis expounded in section 2.1, which prioritizes unbiased estimation, underscores that the ideal CVR loss should entail modeling the fully observed conversion matrix.

In practice terms, conversion labels can only be observed in clicked events but missing in unclicked events, and this pattern is not random [14, 23]. As a result, the CVR prediction task inherently embodies a counterfactual nature. Essentially, during inference, we endeavor to estimate the conversion rates under the hypothetical assumption that all items have been clicked by all users, a scenario that deviates from reality. Thus, the challenge at hand revolves around the effective modeling of the counterfactual space.

To address the aforementioned challenge, we propose the task-level adversarial module. In an effort to integrate information from the impression space into the CVR task, we initially employ a generator designed for exposure events, with the purpose of generating synthetic click embeddings. The objective is to operate under the assumption that all events in the impression space have been clicked, thus achieving the goal of modeling the counterfactual space. The generator can be expressed as follows:

$$\ell_{gt} = \mathbb{E}_{p(z_1')}\left[\left(\mathcal{D}(\mathcal{G}(x_{ctr})) - b_2\right)^2\right], \tag{16}$$

where $p(z_1')$ represents the data distribution in the CTR task, $\mathcal{G}$ denotes the generator, $x_{ctr}$ denotes the CTR prediction embeddings and $b_2$ is the click space label.

Subsequently, we proceed to establish a discriminator to jointly extract task-invariant features for both the CTR and CVR tasks, working in tandem with the generator $\mathcal{G}$. From a task-oriented perspective, the former corresponds to the click task, while the latter pertains to the post-click conversion task, and their invariance revolves around click-related information. From the sample space viewpoint, the overlap between the impression space and the click space encompasses the clicked events. The formulation can be illustrated as follows:

$$\begin{aligned}\ell_{dt} = &\ \mathbb{E}_{p(z_1')}\left[\left(\mathcal{D}(\mathcal{G}(x_{ctr})) - a_2\right)^2\right] \\ &+ \mathbb{E}_{p(z_2')}\left[\left(\mathcal{D}(\mathcal{E}_1(x_{cvr})) - b_2\right)^2\right],\end{aligned} \tag{17}$$

where $p(z_2')$ represents the data distribution in the CVR task, $x_{cvr}$ denotes the CVR prediction embeddings, $a_2$ is the impression space label and $\mathcal{E}_1$ is the CVR encoder.

Nevertheless, the acquired invariant features cannot be straightforwardly integrated into the CVR task. Such integration would introduce elements of click-related information into the conversion task, thereby causing potential bias. Consequently, we introduce a new encoder denoted as $\mathcal{E}_2$ and implement an orthogonal loss $\mathcal{L}_{orthogonal}$ to effectively eliminate any traces of click-related information from the CVR prediction embeddings:

$$\mathcal{L}_{orthogonal} = \frac{\mathcal{E}_1(x_{cvr}) \cdot \mathcal{E}_2(x_{cvr})}{\| \mathcal{E}_1(x_{cvr}) \| \cdot \| \mathcal{E}_2(x_{cvr}) \|}. \tag{18}$$

This approach not only eliminates the inherent click propensity in the CVR task but also introduces exposure-related information acquired under the influence of the orthogonal loss. It's worth noting that we do not substitute the CVR prediction embeddings, which may retain some degree of click-related information, with the learned embeddings. Instead, we employ a concatenation operation. This strategic choice is motivated by the aim of enabling the model to independently learn adaptive weights tailored to the specific demands of the task at hand. To summarize, the task-level loss is presented below:

$$\mathcal{L}_{Task} = \ell_{dt} + \ell_{gt}. \tag{19}$$

### 3.5 Optimization Tasks

As illustrated in Fig. 2, we employ a multi-task framework to concurrently train our proposed model, encompassing three core tasks: CTR, CVR and CTCVR. Note that the estimation of CTCVR is accomplished by computing the product of the estimates obtained for CTR and CVR. In the training of both the CTR and CTCVR tasks,

we employ binary cross-entropy as the objective function, which is formally expressed by the following formulas:

$$\mathcal{L}_{CTR} = \frac{1}{|\mathcal{D}|} \sum_{(u,i)\in\mathcal{D}} o_{u,i} log(\hat{p}_{u,i}) + (1 - o_{u,i})log(1 - \hat{p}_{u,i}), \quad (20)$$

$$\mathcal{L}_{CTCVR} = \frac{1}{|\mathcal{D}|} \sum_{(u,i)\in\mathcal{D}} (t_{u,i}) log(\hat{t}_{u,i}) + (1 - t_{u,i})log(1 - \hat{t}_{u,i}), \quad (21)$$

where $t_{u,i} = o_{u,i} \cdot r_{u,i}$ and $\hat{t}_{u,i} = \hat{p}_{u,i} \cdot \hat{r}_{u,i}$ denote the ground-truth label and the prediction of the CTCVR task, respectively.

To address the challenges posed by the inherent estimation bias (IEB) [3, 22] and potential independent priority (PIP) [22] concerns, we draw inspiration from the setup proposed in [22]. In our approach, we incorporate two equalization factors, i.e., $\lambda_r, \lambda_t$, to regulate the weighting of the CVR task and the CTCVR task during the entire learning process. Overall, the optimization objective for the main branch can be described as follows:

$$\mathcal{L}_{main} = \mathcal{L}_{CTR} + \lambda_r \mathcal{L}_{CVR} + \lambda_t \mathcal{L}_{CTCVR}. \quad (22)$$

## 4 EXPERIMENTS

In this section, we commence by providing a comprehensive introduction to our benchmark datasets and outlining the experimental protocols that we have adopted. Subsequently, we delve into an extensive set of experiments aimed at addressing the following research questions:

- **RQ1:** How much improvement does our proposed framework achieve compared with the state-of-the-art approaches?
- **RQ2:** Are the dual adversarial component and the bias-variance estimator we proposed effective in enhancing performance?
- **RQ3:** Whether the clicked events and unclicked events are blended under the space-level adversarial module and whether the task-invariant features and new unbiased embeddings generated via the orthogonal loss are disentangled under the task-level adversarial module?
- **RQ4:** How do fluctuations in hyperparameters impact the model and does the strategy of balancing bias and variance terms through the bias-variance trade-off estimator prove to be effective?

### 4.1 Dataset

- **Public dataset**: The public dataset Ali-CCP (Alibaba Click and Conversion Prediction) [12] gathered from real-world traffic logs of the recommender system in Taobao. We conduct our experiments following the setup outlined in [22], and the statistics are listed in Table 2.
- **Industrial dataset**: For a more comprehensive and reliable evaluation, we extend our experiments to a large-scale advertising dataset gathered from a real-world system. This dataset is sampled from the offline data logs spanning five consecutive days within the system. The initial four days are dedicated to training, while the last day is reserved for testing. Each sample within this dataset comprises user features, item attributes, and corresponding context information. Please refer to Table 2 for a detailed statistics.

**Table 2: Statistics of the advertising dataset.**

| Dataset | Ali-CCP Dataset | | | Industrial Dataset | | |
|---|---|---|---|---|---|---|
| | #Impression | #Click | #Conversion | #Impression | #Click | #Conversion |
| Training Set | 100K | 4.41K | 0.041K | 11.78M | 3.03M | 0.157M |
| Testing Set | 100K | 4.82K | 0.042K | 3.31M | 0.663M | 55.6K |

### 4.2 Baseline Methods

We compare our proposed framework with the following baselines.

- **ESMM** [12] : It learns CVR utilizing a CTR task and a CTCVR task, which is a non-causal estimator.
- **MTL-IPW** [28] : It implements the IPW estimator within a multi-task framework.
- **MTL-DR** [28] : It deploys the DR estimator within a multi-task framework to debias more robustly.
- **DR-JL** [24] : It jointly trains the prediction model and imputation model in a doubly roubst way.
- **MRDR** [8] : It directly minimizes the variance of the DR estimator to train the imputation model.
- **ESCM$^2$-IPW** [22] : It incorporates the IPW estimator to regularize ESMM's CVR estimation.
- **ESCM$^2$-DR** [22] : It augments ESCM$^2$-IPW with imputation tower and models the CVR risk with the DR estimator.

### 4.3 Implementation Details

In our experiments on the public dataset, we assess the performance of our proposed method on two crucial tasks, i.e., CVR and CTCVR, employing the Area Under the Curve (AUC) [7] metric for evaluation. Following [22]. the feature dimension is set to 5. For experiments on the industrial dataset, we utilize AUC and Precision for evaluating performance. The feature dimension is set to 8. Besides, we employ the *Adam* [10] optimizer with a learning rate of $lr = 1e^{-4}$ and weight decay of $\beta = 1e^{-3}$. The activation functions involved in two discriminators make use of *LeakyReLU* [13], with a negative slope set to 0.2. Codes and datasets will be available on our website.

### 4.4 Overall Performance (RQ1)

To validate the effectiveness of our proposed AECM, we conduct comparative experiments against several baseline methods, which include biased estimators like ESMM and several state-of-the-art unbiased causal estimators. These experiments are conducted on both public dataset and industrial dataset. In the Ali-CCP dataset, we consider two distinct tasks, i.e., CVR and CTCVR, and record data in both click space $O$ and entire impression space $\mathcal{D}$ with the metric AUC for evaluation. Besides, for the industrial scenario, we perform test exclusively in the click space. In contrast to the public dataset where we evaluate utilizing the AUC metric, we introduce the precision as an additional metric for the industrial dataset. The detailed experimental results are documented in Table 3 and Table 4.

As shown in Table 3 and Table 4, we can observe that biased approach ESMM achieves competitive performance on both datasets when compared to MTL-IPW. Specifically, ESMM achieves AUCs of 0.580845 in $O$ and 0.586026 in $\mathcal{D}$ on Ali-CCP, while the MTL-IPW estimator achieve AUCs of 0.573799 and 0.574599, respectively. In addition, several state-of-the-art unbiased causal estimators, including DR-JL, MRDR and ESCM$^2$, demonstrate their strengths gradually. Our proposed adversarial-enhanced causal framework,

**Table 3: Performance comparison for CVR and CTCVR on the public dataset Ali-CCP with the evaluation metric AUC. Note that $O$ and $\mathcal{D}$ represent the results of the click space and the impression space, respectively.**

| | Ali-CCP Dataset | | | |
|---|---|---|---|---|
| Dataset | $O$ | | $\mathcal{D}$ | |
| Model | CVR | CTCVR | CVR | CTCVR |
| ESMM [12] | 0.580845 | 0.571177 | 0.586026 | 0.572875 |
| MTL-IPW [28] | 0.573799 | 0.568533 | 0.574599 | 0.567724 |
| MTL-DR [28] | 0.578944 | 0.572640 | 0.589835 | 0.571525 |
| DR-JL [24] | 0.585492 | 0.584209 | 0.591900 | 0.594481 |
| MRDR [8] | 0.583610 | 0.587182 | 0.591536 | 0.593915 |
| ESCM$^2$-IPW [22] | 0.585219 | 0.579094 | 0.585788 | 0.579141 |
| ESCM$^2$-DR [22] | 0.592165 | 0.585492 | 0.594053 | 0.594660 |
| AECM | **0.610026** | **0.593829** | **0.615735** | **0.596006** |

**Table 4: Performance comparison for CVR on the Industrial dataset in the click space $O$ with the evaluation metric AUC and Precision.**

| | Industrial Dataset | | | |
|---|---|---|---|---|
| Dataset | CVR | | CTCVR | |
| Model | AUC | Precision | AUC | Precision |
| ESMM [12] | 0.808093 | 0.447450 | 0.807907 | 0.452819 |
| MTL-IPW [28] | 0.805792 | 0.476168 | 0.802976 | 0.483986 |
| MTL-DR [28] | 0.808317 | 0.479965 | 0.805634 | 0.487812 |
| DR-JL [24] | 0.812932 | 0.455961 | 0.808873 | 0.458873 |
| MRDR [8] | 0.813605 | 0.473534 | 0.812628 | 0.471639 |
| ESCM$^2$-IPW [22] | 0.816218 | 0.477207 | 0.816101 | 0.479079 |
| ESCM$^2$-DR [22] | 0.818261 | 0.485235 | 0.818120 | 0.488576 |
| AECM | **0.824042** | **0.502605** | **0.824023** | **0.503134** |

AECM, surpasses these suboptimal approaches with significant improvements. On Ali-CCP, AECM achieves AUCs of 0.610026 in $O$ and 0.615735 in $\mathcal{D}$, representing a substantial increase compared to ESCM$^2$-DR by 3.01% and 3.6%, respectively. On the industrial dataset, AECM achieves an AUC of 0.824042 and a precision of 0.502605, translating to boosts of 0.706% and 3.57%, respectively.

## 4.5 Ablation Study (RQ2)

As depicted in Fig. 2, we introduce a dual adversarial component consisting of the space-level module and the task-level module to effectively eliminate click propensity within our framework. In addition, we develop the unbiased bias-variance balance estimator to enhance the robustness of our framework. To investigate the individual contributions of these components, we conduct an ablation study, and the results are summarized in Table 5. We systematically get rid of the space-level module, the task-level module, and the bias-variance balance estimator to assess their impact. Taking a holistic view, each of the proposed modules plays a crucial role in improving AUC. Specifically, on the public dataset Ali-CCP, the

**Table 5: Ablation Study on the public dataset and the industrial dataset, which exclude the space-level adversarial module, the task-level adversarial module and the bias-variance estimator, respectively. Note that the results on Ali-CCP and industrial dataset are observerd in $\mathcal{D}$ and $O$, respectively.**

| Method | | Ali-CCP Dataset | | Industrial Dataset | |
|---|---|---|---|---|---|
| | | CVR | CTCVR | CVR | CTCVR |
| w/o | Space-level | 0.597207 | 0.572899 | 0.821755 | 0.821756 |
| | Task-level | 0.611296 | 0.589949 | 0.822402 | 0.822381 |
| | Bias-Variance | 0.605161 | 0.586743 | 0.820696 | 0.820662 |
| ours | AECM | 0.615735 | 0.596006 | 0.824042 | 0.824023 |

space-level adversarial module provides the most significant performance boost. When this module is removed, there is a noteworthy decline in CVR task AUC, decreasing from 0.615735 to 0.597207. Conversely, on the industrial dataset, the estimator demonstrates its significance in enhancing unbiased CVR estimation. Removal of the estimator results in a performance drop from 0.824042 to 0.820696 in the CVR task. These findings emphasize the importance of each component in our framework.

## 4.6 Feature Visualization (RQ3)

The space-level and task-level adversarial modules, as introduced in Section 3.3 and Section 3.4, respectively, are designed to mitigate the influence of click propensity, which can introduce interference in CVR estimation. The space-level module employs $\mathcal{L}_{Space}$ to align the distribution of clicked and unclicked events, effectively merging them into the same feature space. On the other hand, the task-level module generates disentangled CVR prediction embeddings through the combined impact of $\mathcal{L}_{Task}$ and the orthogonal loss $\mathcal{L}_{orthogonal}$. To validate the effectiveness of our dual adversarial modules, specifically, whether they successfully produce mixed embeddings learned in a domain-adversarial fashion at the space-level and disentangled embeddings at the task-level, we utilize a nonlinear dimensionality reduction technique, t-SNE [21], to transform the high-dimensional features into two-dimensional representations. We then visualize their distributions on a flat map.

Fig. 4(a) and Fig. 4(c) display the characteristic distribution of embeddings at the space-level for the public dataset and industrial dataset, respectively. Similarly, Fig. 4(b) and Fig. 4(d) illustrate the distribution at the task-level for the public dataset and industrial dataset, respectively. The visualizations in the graphs clearly indicate that clicked events (green dots) and unclicked events (cherry blossom dots) are fully integrated into the same feature space, as they appear intermingled. Moreover, the click-related noise (deep blue) and the unbiased prediction embedding (light blue) are distinctly separated and disentangled from each other. These observations provide a degree of confirmation for the effectiveness of our proposed framework and the validity of our experimental results.

## 4.7 Parameter Analysis (RQ4)

We have three critical hyperparameters that influence the performance of the AECM model, namely, $\lambda_r$ and $\lambda_t$ in Equation (22), as well as $\lambda_\tau$ in Equation (10). The experimental results are presented in Fig. 5. (a) and (c) present the values for CVR and CTCVR on the

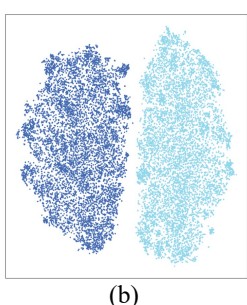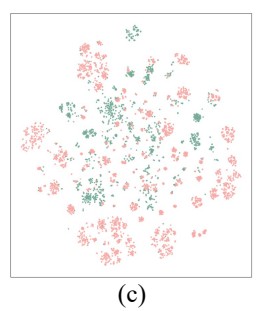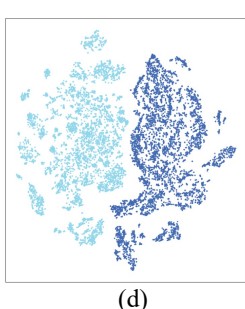

Figure 4: The t-SNE visualization of feature distributions for both the space-level and task-level modules. (a) and (b) are on Ali-CCP, with cherry blossom color representing the exposure events and green representing the click events; light blue represents embeddings without click-related information learned under the orthogonal loss. (c) and (d) are the corresponding illustrations on the Industrial dataset.

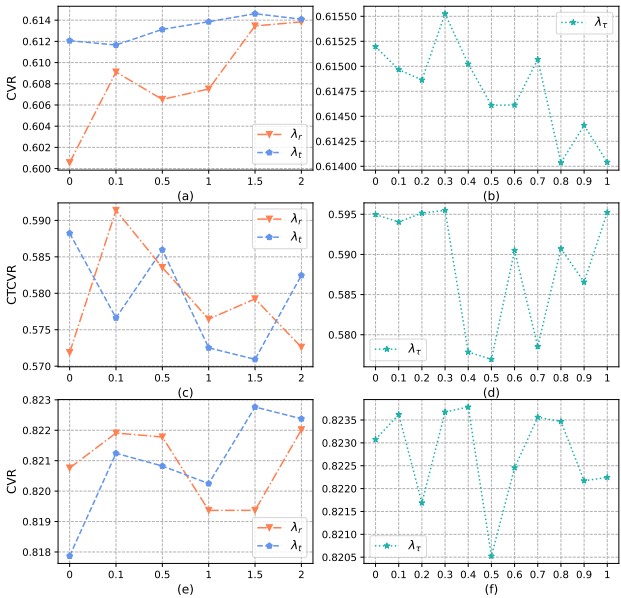

Figure 5: Parameter study of $\lambda_r, \lambda_t, \lambda_\tau$ on (a-b): CVR estimation on Ali-CCP; (c-d): CTCVR estimation on Ali-CCP; (e-f): CVR estimation on Industrial dataset.

Ali-CCP dataset. (e) provides the corresponding data for the CVR task on the industrial dataset. In (b), (d), and (f), we delve into a detailed analysis of the impact of the factor $\lambda_\tau$. In particular, we vary $\lambda_r$ and $\lambda_t$ in the range [0, 2]. As we increment $\lambda_r$, we consistently observe improvements in the performance of CVR estimations. For instance, the AUC of CVR shows a noticeable enhancement, rising from 0.6 when $\lambda_r = 0$ to approximately 0.614 when $\lambda_r = 2$. However, it's essential to note that the AUC of CTCVR decreases with higher values of $\lambda_r$. Therefore, we strike a balance between these two tasks by carefully adjusting the parameter $\lambda_t$. We ultimately determine that the values for $\lambda_r$ and $\lambda_t$ are 2 and 1.5 respectively when the optimal AUC is obtained.

For bias-variance balance factor $\lambda_\tau$, we vary it within the range [0, 1] with a step size of 0.1. Fig. 5(b) and Fig. 5(d) illustrate that when $\lambda_\tau = 0.3$, both CVR and CTCVR tasks obtain the optimal AUC on the Ali-CCP dataset. On the industrial dataset, the best performance is achieved when $\lambda_\tau = 0.4$. Consequently, we set $\lambda_\tau$ to 0.3 for the Ali-CCP dataset and 0.4 for the industrial dataset.

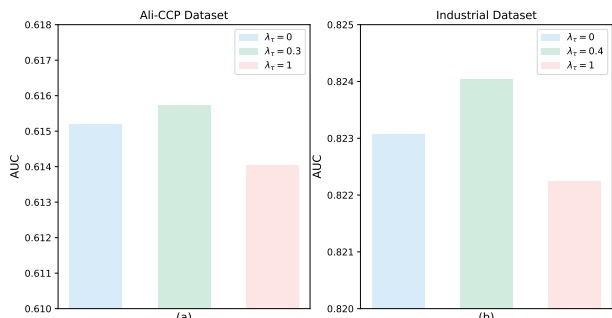

Figure 6: The effectiveness of the balance factor $\lambda_\tau$.

Besides, this finding demonstrates its superiority over situations where it is constrained to either 0 or 1, indirectly suggesting that the $\lambda_\tau$ factor in Eq. (10) plays a pivotal role in achieving a balance between the bias and variance terms, as illustrated in Fig. 6.

## 5 CONCLUSION

In this paper, we address the pervasive challenge of selection bias in post-click conversion rate (CVR) estimation. Our approach commences with a comprehensive theoretical analysis that delves into the deviations inherent in DR estimators and their implications for the generalization bound. Building upon this foundational theory, we propose a novel causal estimator, taking into account the bias-variance trade-off perspective. Subsequently, we provide a novel dual adversarial module designed to mitigate the impact of click propensity in the CVR task. This dual module operates on two distinctive levels: the space level and the task level. On the space level, it seeks to align the representations of both clicked and unclicked events in the impression space. Concurrently, on the task level, the focus shifts towards learning the invariance, essentially the click-related information shared between the CTR and CVR tasks. This click-related information is effectively removed from CVR embeddings through an orthogonal loss. In the final phase of our study, we conduct extensive experiments utilizing both public dataset and industrial dataset to validate the effectiveness of our proposed model. The empirical findings convincingly demonstrate the superiority of our approach in addressing the challenges arising from selection bias and data sparsity. In forthcoming research endeavors, we intend to further refine and enhance our model, thus contributing to the ongoing advancement of research in the domain of sample selection bias.

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
