# OpenReview forum: "Adversarial-Enhanced Causal Multi-Task Framework for Debiasing Post-Click Conversion Rate Estimation"
_ACM.org/TheWebConf/2024/Conference — TheWebConf24_

### Official Review · Reviewer_tGib · 2023-11-02

**Novelty:** 3
**Technical Quality:** 5

**Review:**

**Summary**


The paper addresses practical challenges in post-click conversion rate (CVR) prediction in affiliate advertising due to data sparsity and sample selection bias. Recognizing the limitations of current Doubly Robust (DR) estimators, particularly their inaccuracy in imputed errors, the authors propose an Adversarial-Enhanced Causal Multitask (AECM) framework. This approach integrates a novel causal estimator and a dual adversarial component to mitigate bias and variance in CVR predictions. The causal estimator optimizes the imputation model, while the adversarial component addresses selection bias by aligning distributions in the click and impression spaces and denoising click propensities. The paper claims effectiveness based on extensive experiments and also promises to release a valuable dataset for selection bias research in advertising.

**Strengths**

- The paper tackles the practically important problem of Post-Click Conversion Rate prediction taking selection bias and data sparsity into consideration

- The paper is well written and the arguments are easy to understand. In particular, Figures 1 and 3 are quite useful to grasp the key concept of the problem


- The paper's findings are tested through extensive experiments on real-world datasets. Experiment design and choice of baselines look sufficient, and the provided results are promising and show the advantages of the proposed debiasing method


- Release of a new large-scale dataset is a big plus to the relevant community

**Weaknesses**

- Even though CVR prediction under selection bias and data sparsity is a practically relevant and important area of research, it is extremely dense (there are many baselines in the experiments indeed), so proposing some extensions of DR might be considered a marginal contribution


- Theorems 1 to 3 are well-known results, and thus I was not sure how useful it is to consume about a page to state and prove these known results.


- There are no theoretical backgrounds on come components of the proposed method


- Even though the paper performs offline experiments on a couple of real-world datasets, it fails to perform online A/B test; so it might be risky to be overly optimistic about the empirical results of the paper until the proposed approach is verified in an online environment and provides some tangible business benefits

**Questions:**

- I was not sure why it is necessary to re-prove Theorems 1 and 2,


- Is it possible to provide some theoretical background for every component (or module) of the proposed method?

**Reviewer Confidence:**

3: The reviewer is confident but not certain that the evaluation is correct

**Scope:**

4: The work is relevant to the Web and to the track, and is of broad interest to the community

---

### Official Review · Reviewer_X4ai · 2023-11-22

**Novelty:** 6
**Technical Quality:** 4

**Review:**

The manuscript integrates adversarial learning and causal multi-task learning, and includes an analysis of the bias and variance inherent in DR estimators.
The paper also presents a novel dual adversarial module that operates on both the space and task level to denoise the click propensity and capture counterfactual embeddings to optimize the CVR task.

Advantages:
1. Propose a causal multi-task framework, named AECM, to deal with the selection bias issue for post-click conversion rate estimation.
2. Introduce a theoretical analysis of the bias and variance inherent in DR estimators
3. Design a novel causal estimator to minimize both bias and variance from the bias-variance trade-off perspective.
4. Reasonable model performance improvement

Shortcomings:
1. No related work
2. The author has compressed an extensive amount of information within the confines of limited pages, making it challenging to grasp all the details of this article.
3. Clear explanations and step-by-step derivations would be beneficial for better understanding.

**Questions:**

[Regarding Bias and Variance]
1. How are bias term and variance term obtained in the generalization bound in Theorem 3?
2. What do bias and variance terms refer to in such a multi-task recommendation scenario? What specific impact do they have on the model?
3. Detailed reasoning and/or derivation processes from EQ.(7) and (8) to generalization bound, and its extension to Eq.10 are necessary for understanding the optimization. Please provide a coherent and well-justified explanation for the transitions between these three sections.

[Regarding trade-off in Estimator]
4. Trade-off implies finding a balance between two variables. However, it seems that bias and variance in Theorem 3 and Equation 10 do not follow this pattern; low bias and low variance are not mutually exclusive. Why not simultaneously minimize bias and variance, but choose to trade-off between the two terms?
5. In line 561 on page 5, the author proposes, "This approach not only eliminates the inherent click propensity … under the influence of the orthogonal loss." What is the basis for this statement?

[Regarding experiments]
6. In the experimental section, significance testing is necessary.

**Reviewer Confidence:**

4: The reviewer is certain that the evaluation is correct and very familiar with the relevant literature

**Scope:**

4: The work is relevant to the Web and to the track, and is of broad interest to the community

---

### Official Review · Reviewer_5oBG · 2023-11-23

**Novelty:** 4
**Technical Quality:** 5

**Review:**

This study works on debiased post-click conversion rate estimation from a causal perspective. The authors extend doubly robust estimator by taking bias-variance balance into account. Meanwhile, the authors incorporate a dual adversarial module in an affort to address improve the debiasing capacity. Empirical evaluations, using a public and an industrial dataset, compare this approach against 7 baselines.

**pros**

1. the authors have discussed a strong motivation, identifying gaps in previous studies and how they propose to handle them.
2. the bias-variance trade-off estimator is built upon theoretically informed generalization bound.

**cons**

1. I raise my concern about the scalability and complexity of this model in practical implementation due to dual adversarial modules. It is often the case that adversarial training is not easy to go with, not to mention a dual level. Also, it appears the tradeoff hyperparameters $\lambda_r$ and $\lambda_t$ need to be determined empirically, varying from dataset to dataset.
2. lack of a comprehensive discussion of related studies
3. I observe all the three model variants seem to have negligible impact on industrial dataset performance. Also, the reported numerical results do not include sinigificance test or std.

**Questions:**

please kindly address the concerns as pointed out in the **cons** section.

**Ethics Review Description:**

no ethical concerns were identified, but the empirical study includes a private dataset, which might be of concerns in this context.

**Reviewer Confidence:**

3: The reviewer is confident but not certain that the evaluation is correct

**Scope:**

4: The work is relevant to the Web and to the track, and is of broad interest to the community

---

### Official Review · Reviewer_x9hu · 2023-11-24

**Novelty:** 4
**Technical Quality:** 4

**Review:**

This paper tackles the prevalent issue of selection bias in post-click conversion rate (CVR) estimation. It begins with an in-depth theoretical analysis of deviations in DR estimators and their impact on generalization bounds. The authors then introduce a novel causal estimator that considers the bias-variance trade-off. To address click propensity in CVR estimation, they propose a dual adversarial module functioning at two levels: aligning clicked and unclicked event representations in the impression space, and learning invariant click-related information across CTR and CVR tasks, which is then removed from CVR embeddings via an orthogonal loss. The model's effectiveness is validated through extensive experiments with public and industrial datasets, showing its superiority in overcoming selection bias and data sparsity challenges. Future work aims to refine the model further, contributing to research in the field of sample selection bias.

**Questions:**

What is the limitation of  your work?
Could you do more experiments on different dataset？

**Reviewer Confidence:**

1: The reviewer's evaluation is an educated guess

**Scope:**

4: The work is relevant to the Web and to the track, and is of broad interest to the community

---

### Official Review · Reviewer_qGcA · 2023-11-29

**Novelty:** 6
**Technical Quality:** 6

**Review:**

**Summary**

This paper proposes a multi-task learning method named AECM to address selection bias problem in recommender systems. The proposed method uses adversarial learning technic in space-level adversarial module and task-level adversatial module to enhance the debiasing performance. The real-world experiment results verify the effectiveness of the proposed method.

**Pros**

- The research question is interesting and important.
- This paper is well-organized.
- The real-world experiments verify the effectiveness of the proposed method.

**Cons**

My major concern is the lack of methodology novelty and many related work. I am willing to change my score if the authors can address my following concerns.

- Methodology novelty. The adversarial balancing for debiased recommendation is proposed by [1] and the adaptive bias-variance trade-off  for debiased CTCVR is proposed by [2], which studied the exact same problem compared to this paper. The authors should discuss the relation and the difference in their manuscript.

- Lack of related work. There are a branch of debiasing papers should be discussed and compared as the baselines in this paper, such as information bottleneck based methods [3, 4], enhanced doubly robust methods [5-9], multiple robust method [10], debiasing methods with a few unbiased ratings [11], and distribution calibration method [12, 13]. All these papers focus on the same problem setup.

- Theorems 1-3 are well-known results: the bias and variance of DR and MRDR have already been widely studied.

- The authors should provide error bars and conduct hypothesis tests to verify the statistical significance of the experimental results.

- Some typos exist, e.g., "Following [22]." in line 665.

**References**

[1] Mengyue Yang, Guohao Cai, Furui Liu, Jiarui Jin, Zhenhua Dong, Xiuqiang He, Jianye Hao, Weiqi Shao, Jun Wang, and Xu Chen. 2023. Debiased recommendation with user feature balancing. In ACM Transactions on Information Systems.

[2] Quanyu Dai, Haoxuan Li, Peng Wu, Zhenhua Dong, Xiao-Hua Zhou, Rui Zhang, Xiuqiang He, Rui Zhang, and Jie Sun. 2022. A Generalized Doubly Robust Learning Framework for Debiasing Post-Click Conversion Rate Prediction. In KDD.

[3] Zifeng Wang, Xi Chen, Rui Wen, Shao-Lun Huang, Ercan E Kuruoglu, and Yefeng Zheng. 2020. Information Theoretic Counterfactual Learning from Missing-Not-At-Random Feedback. In NeurIPS.

[4] Dugang Liu, Pengxiang Cheng, Hong Zhu, Zhenhua Dong, Xiuqiang He, Weike Pan, and Zhong Ming. 2021. Mitigating Confounding Bias in Recommendation via Information Bottleneck. In RecSys.

[5] Yuta Saito. 2020. Doubly robust estimator for ranking metrics with post-click conversions. In RecSys.

[6] Haoxuan Li, Yan Lyu, Chunyuan Zheng, and Peng Wu. 2023. TDR-CL: Targeted Doubly Robust Collaborative Learning for Debiased Recommendations. In ICLR.

[7] Haoxuan Li, Chunyuan Zheng, and Peng Wu. 2023. StableDR: Stabilized Doubly Robust Learning for Recommendation on Data Missing Not at Random. In ICLR.

[8] Haoxuan Li, Yanghao Xiao, Chunyuan Zheng, Peng Wu, and Peng Cui. 2023. Propensity Matters: Measuring and Enhancing Balancing for Recommendation. In ICML.

[9] Zijie Song, Jiawei Chen, Sheng Zhou, QiHao Shi, Yan Feng, Chun Chen, and Can Wang. 2023. CDR: Conservative Doubly Robust Learning for Debiased Recommendation. In CIKM.

[10] Haoxuan Li, Quanyu Dai, Yuru Li, Yan Lyu, Zhenhua Dong, Xiao-Hua Zhou, and Peng Wu. 2023. Multiple Robust Learning for Recommendation. In AAAI.

[11] Liu, Haochen, Da Tang, Ji Yang, Xiangyu Zhao, Hui Liu, Jiliang Tang, and Youlong Cheng. 2022. Rating distribution calibration for selection bias mitigation in recommendations. In WWW.

[12] Haoxuan Li, Yanghao Xiao, Chunyuan Zheng, and Peng Wu. 2023. Balancing unobserved confounding with a few unbiased ratings in debiased recommendations. In WWW.

[13] Jiawei Chen, Hande Dong, Yang Qiu, Xiangnan He, Xin Xin, Liang Chen, Guli Lin, and Keping Yang. 2021. AutoDebias: Learning to Debias for Recommendation. In SIGIR.

**Questions:**

Please refer to the **Cons** part for the questions.

**Reviewer Confidence:**

4: The reviewer is certain that the evaluation is correct and very familiar with the relevant literature

**Scope:**

4: The work is relevant to the Web and to the track, and is of broad interest to the community

---

### Decision · Program_Chairs · 2024-01-22

**Decision:**

Accept

**Comment:**

This a technically well-executed paper with a novel methodology for post-click conversion rate (CVR) estimation that includes industrial-scale experimentation and encouraging live A/B test results as reported in the rebuttal. Reading through the author response to the reviewers, all major concerns appear to have been addressed (as acknowledged by many reviewers, though some did not respond) and overall reviewer support ranges from neutral to positive for acceptance. Given the combined technical and empirical contributions of the work as confirmed by my own read of the paper, I recommend it for acceptance.

 One reviewer has asked for the rebuttal discussion to be added to the next revision of the paper and I generally encourage the authors to add useful clarifications from their discussion to on the next revision.